# Benchmarking Speech-Driven Gesture Generation Models for Generalization to Unseen Voices and Noisy Environments

Johsac I. G. Sanchez
Kevin Inofuente-Colque
j216401@dac.unicamp.br
k272316@dac.unicamp.br
Department of Computer Engineering
and Automation, FEEC - Unicamp
Campinas, SP, Brazil

Leonardo B. de M. M. Marques*
lmenezes@cpqd.com.br
CPqD
Campinas, SP, Brazil

Paula D. P. Costa
Rodolfo L. Tonoli
paulad@unicamp.com
r105652@dac.unicamp.br
Department of Computer Engineering
and Automation, FEEC - Unicamp
Campinas, SP, Brazil

## Abstract

Speech-driven gesture generation models enhance robot gestures and control avatars in virtual environments by synchronizing gestures with speech prosody. However, state-of-the-art models are trained on a limited number of speakers, with audios typically recorded in controlled conditions, potentially resulting in poor generalization to new voices and noisy environments. This paper presents a robust evaluation method for speech-driven gesture generation models against unseen voices and varying noise levels. We utilize a voice conversion model to produce synthetic speech that maintains prosodic features, ensuring a thorough test of the model's generalization capabilities. Additionally, we introduce a controlled synthetic noisy dataset to evaluate model performance under different noise conditions. This methodology establishes a comprehensive framework for robustness evaluation in speech-to-gesture synthesis benchmarks. Applying this approach to the state-of-the-art DiffuseStyleGesture+ model reveals a slight performance degradation with diverse voices and increased background noise. Our findings emphasize the need for models that can generalize better to real-world conditions, ensuring reliable performance in varied acoustic scenarios.

## CCS Concepts

• **Computing methodologies** → **Machine learning**; • **Human-centered computing** → *Human-computer interaction (HCI)*.

## Keywords

Gesture generation, robustness, voice conversion, noise

**ACM Reference Format:**
Johsac I. G. Sanchez, Kevin Inofuente-Colque, Leonardo B. de M. M. Marques, Paula D. P. Costa, and Rodolfo L. Tonoli. 2024. Benchmarking Speech-Driven Gesture Generation Models for Generalization to Unseen Voices and Noisy Environments. In *INTERNATIONAL CONFERENCE ON MULTIMODAL INTERACTION (ICMI Companion '24), November 4–8, 2024, San Jose, Costa Rica.* ACM, New York, NY, USA, 5 pages. https://doi.org/10.1145/3686215.3688823

*Also with Department of Computer Engineering and Automation, FEEC - Unicamp.

## 1 Introduction

Multispeaker gesture generation models from speech have become increasingly significant in the realm of virtual reality, enabling a plethora of applications. For example, users can control avatars within the metaverse, participate in virtual games, or attend virtual meetings, all with gestures dynamically generated from their spoken words. This technology is particularly advantageous in virtual production, as it facilitates the creation of characters without the need for motion capture. In this context, the capacity of these models to adapt to previously unseen conditions, such as new voices or noisy environments, is a key challenge towards more personalized virtual experiences since it can ensure that the avatars' gestures align closely with the user's personal style, enhancing virtual characters' realism and individuality.

Previous research has explored generating gestures specific to different speakers, successfully capturing their unique styles [1, 5, 22]. Additionally, style-transfer techniques have shown promising results in generating gestures for unseen styles [3, 4]. Diffusion-based models gained a lot of attention in recent works in gesture generation [15, 18]. However, a significant gap remains in generalizing gesture generation models to unseen voices. Despite claims of handling synthesized audio or being speaker-agnostic [4, 22], a comprehensive evaluation across diverse unseen voices is still lacking.

Moreover, applications such as virtual meetings in busy offices or gaming in public places often involve significant background noise. For these applications, speech-driven gesture generation systems should generalize not only to different voices but also to various environmental noise conditions. This is challenging due to the limited datasets with multiple speakers in diverse conditions and styles. Also, most datasets are recorded in controlled environments, such as motion capture or video studios, which do not reflect real-world settings.

On the other hand, evaluating gesture generation models is difficult due to the variability of gestures and subjective aspects of human motion quality. The gold standard is perceptual evaluations for human-likeness and speech appropriateness [9]. However, user studies are costly and time-consuming, making including variables like unseen voices and environmental noise during development impractical. Thus, an objective benchmark for these aspects is highly

valuable. Currently, no benchmarks exist for assessing the robustness of speech-driven gesture generation models to unseen voices and environmental noise.

To address this gap, this work presents three main contributions: (1) We propose a method to evaluate the robustness of speech-driven gesture generation models to unseen voices using voice conversion (VC) speech synthesis models. This approach enables controlled experiments without the need for new studio recordings. (2) We introduce TWH-Party, a new dataset derived from the Talking with Hands (TWH) dataset [10], which includes speech stimuli with varying levels of background noise. (3) We benchmark the state-of-the-art model DiffuseStyleGesture+ [21] against conditions not encountered during its training and provide a detailed discussion of the results.

## 2 Proposed Method

In this section, we describe the key components of our proposed method to establish a benchmark for assessing robustness to unseen conditions in speech-to-gesture synthesis models. First, for any trained model, we propose to evaluate it against unseen voices generated synthetically by a voice conversion system rather than using new studio recordings. Section 2.1 explains how state-of-the-art VC systems can be utilized as experimental tools for controlled experiments. Second, to assess the robustness of the models to degraded input speech, we constructed a dataset of speech stimuli with varying levels of background noise derived from the TWH dataset, a large-scale public dataset of synchronized motion and audio (Section 2.2). Finally, we propose evaluating the models under unseen conditions during the training stage using objective metrics such as Frechét Gesture Distance (FGD) (Section 2.3).

### 2.1 Voice Conversion

Voice conversion (VC) alters a speaker's voice in a given input audio to that of another speaker while preserving linguistic, paralinguistic (e.g., pitch, volume, intonation, rhythm), and non-verbal (e.g., breathing, laughing, crying) information [12]. VC has been used in benchmarks like the ASVspoof challenge for spoofed and deepfake speech detection, where it created voice deepfake attacks to test participants' solutions detection abilities [19].

Standard VC models are enhanced using fundamental frequency ($F_0$) as a conditioning factor. It allows not only to achieve disentanglement between input $F_0$ and speaker timbre information but also to make the output $F_0$ controllable with the modification of the input curve [13]. This approach is particularly successful for singing voice conversion (SVC) models [11]. In this work, we propose to use a state-of-the-art open-source SVC model due to its ability to synthesize expressive speech, a key factor in addressing the expressiveness of speech-synchronized gesture datasets [6].

In particular, our experimental setup used So-VITS-SVC model[1] to convert the existing audios to unseen voices. Its architecture includes four pre-trained audio encoders: a timbre encoder for speaker representations [17], a Whisper encoder for linguistic content [14], a soft HUBERT encoder for prosody [16], and a CREPE model for fundamental frequency [7]. These representations are

processed by a normalizing flow-based decoder, trained to disentangle speaker information so that different timbres can be applied during inference.

### 2.2 Noise Corruption

To assess the robustness of the models to degraded input speech, we propose using a dataset of speech stimuli with varying levels of background noise. We built this dataset by extracting the audio from the 17-speaker TWH dataset subset used in the GENEA Challenge 2023 [9] and synthesizing new audios with different degrees of environmental noise. We named this new dataset "TWH-party".

TWH-party dataset was created by corrupting the original TWH dataset with convolutional reverb randomly extracted from the Room Impulse Response and Noise Database [8], to add an ambient factor, and also with added environmental background, transient, and speech noises, using randomly selected audios from the LibriParty[2] dataset. The LibriParty[2] dataset is a synthetic augmentation of the LibriSpeech dataset with reverb, background and transient noises to simulate a cocktail-party/meeting scenario.

To evaluate the model's robustness in scenarios with different levels of background noise, we parameterize the TWH-party generation scripts. These parameters control the volume of input audio from the TWH to be corrupted (S), the amount of reverb (R), the number of corruption audios added from the Libriparty (N), and their combined volume level (V). We made available the TWH-party as well as its generation scripts to encourage other researchers to evaluate their systems in these settings.

### 2.3 Comparing Generated Motion Sequences

We propose to adopt the Frechét Gesture Distance (FGD) [22] and Mean Squared Error (MSE) to compare the generated motion sequences. The FGD is based on an unsupervised motion feature extractor, and it represents the distance between the Gaussian mean and covariance of the extracted motion features of two sets of gestures, $X_1$ and $X_2$. The FGD is defined as follows:

$$FGD(X_1, X_2) = \|\mu_1 - \mu_2\|^2 + \text{Tr}(\Sigma_1 + \Sigma_2 - 2(\Sigma_1\Sigma_2)^{1/2}), \quad (1)$$

where $\mu_i$ and $\Sigma_i$ are the first and second moments of the latent feature distribution $Z_i$ of gestures $X_i$. In our experimental setup, we trained an autoencoder using the train set of the GENEA Challenge 2023 in sequences of joints' global positions and used the encoder part of the network as the feature extractor.

Additionally, MSE is used to measure the average squared differences between two sets of gestures. It is defined as:

$$MSE = \frac{1}{n} \sum_{i=1}^{n} (X_1 - X_2)^2, \quad (2)$$

where $X_1$ and $X_2$ represent the generated gesture positions.

## 3 Experimental Setup

This section describes our experimental approach to benchmark the state-of-the-art DiffuseStyleGesture+ model against unseen conditions during the training stage.Our code is publicly available at https://github.com/AI-Unicamp/Benchmarking-SDGG-Models

---

[1]Available at https://github.com/PlayVoice/whisper-vits-svc

[2]Available at https://github.com/speechbrain/speechbrain/tree/develop/recipes/LibriParty/generate_dataset

## 3.1 Model of Study: DiffuseStyleGesture+

DiffuseStyleGesture+ model is considered state-of-the-art in speech-driven gesture generation and was one of the top models in the GE-NEA Challenge 2023 [9], also winning the reproducibility award in the challenge. DiffuseStyleGesture+ is a multimodal gesture generation model designed to automatically generate co-speech gestures synchronized with speech [21]. The model integrates data sources like audio, text, and speaker characteristics, mapping these modalities to a latent space. Subsequently, through a diffusion model, DiffuseStyleGesture+ proved to be one of the best evaluated in the challenge to process and produce gestures synchronized with speech.

DiffuseStyleGesture+ excels in integrating various speech features, including (MFCCs, mel-spectrogram, pitch, energy, WavLM [2] representations, and onsets), to generate highly accurate and contextualized gestures [20, 23]. The model processes both local and global audio features, incorporating semantic information from text and speaker-specific data. This multimodal integration ensures that generated gestures accurately reflect speech intonation, rhythm, and content, adapting to both the main speaker and interlocutors in complex conversations [18].

The ability to capture diverse speech features is crucial for producing natural and expressive gestures, enhancing interactivity and fluency. However, since features like mel-spectrogram, MFCCs, and WavLM [2] representations used in the model contain speaker timbre information, the model is inherently conditioned on timbral information of the speakers present in training, thus with the potential to compromise its generalization to unseen voices.

## 3.2 Robustness to Unseen Voices

First, we evaluated the model using input speech audio with a voice different from the input Speaker ID and unseen during training. Our experiment used the DiffuseStyleGesture+ model, pre-trained with the TWH subset of GENEA Challenge 2023. The TWH subset used in the GENEA Challenge 2023 includes 17 speakers. We focused on Speaker 1, who represents 43% of the training set and 80% of the test set (the other 16 speakers individually represent less than 5% of the training set). We applied the SVC model to convert the test utterances of the Speaker 1 to several different out-of-distribution voices.

As the VC model, we used an implementation of the So-VITS-SVC model that is pre-trained on a large multi-speaker speech and singing voice corpus. This model is fine-tuned with the learning rate set to a tenth of the original for 100 epochs on the English partition of the Emotional Speech Dataset (ESD), which consisted of 29 hours of speech data divided across 10 gender-balanced speakers, and in 5 styles: "angry", "happy", "neutral", "sad", and "surprise" [24]. This step yields a VC model that can convert any input speech into the voice of any ESD speaker.

The audios from the TWH's Speaker 1 were converted to the voices of some speakers recorded by the ESD dataset. Aiming to ensure that a greater diversity of unseen voices is evaluated, four speakers of the ESD with different genders and vocal ranges were selected:

- Speaker 12, a male voice with the highest mean pitch (160 Hz) across all ESD male speakers (Condition **Man High Pitch**);
- Speaker 20, a male voice with the lowest mean pitch (115 Hz) across all ESD male speakers (Condition **Man Low Pitch**);
- Speaker 18, a female voice with the highest mean pitch (239 Hz) across all ESD female speakers (Condition **Woman High Pitch**);
- Speaker 19, a female voice with the lowest mean pitch (150 Hz) across all ESD female speakers (Condition **Woman Low Pitch**).

To appropriately convert the utterances from the Speaker 1 of the TWH dataset to all the conditions described above, each possessing different vocal ranges, a fundamental frequency matching algorithm was applied during the conversions. This algorithm consists in transposing the input pitch to match the vocal range of the output speaker. For example, when converting an utterance from the Speaker 1 of the TWH (a female voice) to the Speaker 12 of the ESD (a male voice), the extracted input pitch curve from the Speaker 1 was transposed down 4 semitones so that the generated converted output speech sits on the usual vocal range of the Speaker 12, which is in a lower register. An upwards transposition was used to convert audio from a lower register speaker to a higher register speaker.

After converting the test set partition of Speaker 1 to each of the four voices (conditions), these new voice sets were used as inputs to the DiffuseStyleGesture+ algorithm, resulting in the respective generated movements. Consequently, with the generated movements from the test set of Speaker 1 and the generated movements from the new four voice sets, the next step was to obtain the representations of 3D rotations and positions. These position representations were subsequently used for calculating the FGD metric. For this FGD calculation, pairwise comparisons were made, that is, comparisons between the gesture set from the test data of Speaker 1 and each of the gesture sets from the four voices (Man High Pitch, Woman Low Pitch, Woman High Pitch, Man Low Pitch). Additionally, the MSE calculation was performed. The results of the comparisons for both FGD and MSE are provided in Table 1 left part in Section 4.

## 3.3 Robustness to Environmental Noises

In Section 2.2, we introduced TWH-party; therefore, in this section, we explain how this dataset is applied to the current proposal. To begin with, the TWH-party dataset contains three subsets of audios, where each subset is a copy of the TWH test audio dataset, with the difference that each subset has been added with a different level of noise. Consequently, three levels of noisy dataset were considered, named conditions: low, medium, and high. These were constructed with the following set of parameters, which are also explained in Section 2.2:

- Condition **Low Noise** : {S = -3dB, R = 0.45, N = 6, V = 0.10}.
- Condition **Medium Noise** : {S= -4dB, R= 0.55, N= 8, V= 0.12}.
- Condition **High Noise** : {S = -5dB, R = 0.65, N = 10, V = 0.15}.

Therefore, the GENEA Test Dataset can be considered as "clean from environmental noise" to a certain extent, as it only contains the natural noise of the speakers talking during the interview, without

**Table 1: Results of the evaluation of the Frechét Gesture Distance (FGD) and Mean Squared Error (MSE) metrics when evaluated under different voice conditions and noise levels. Lower values in FGD and MSE indicate better performance.**

| Metric | Man High Pitch | Woman Low Pitch | Woman High Pitch | Man Low Pitch | Low Noise | Medium Noise | High Noise |
|--------|------|------|------|------|------|------|------|
| FGD | **15.74** | 20.36 | 28.08 | 50.25 | **10.56** | 14.23 | 17.59 |
| MSE | **2.38** | 2.51 | 2.61 | 3.06 | **1.82** | 1.95 | 2.01 |

any deliberately added noise using tools or algorithms. On the other hand, the three new datasets (TWH-party) have deliberately added noise at different levels to test the gesture generation model's robustness.

TWH-party served as new inputs to the DiffuseStyleGesture+ algorithm, resulting in the respective generated movements, that is, three sets of gestures. Subsequently, the representations of 3D rotations and positions for each of the three gesture sets were obtained. Finally, the position representations were used to calculate the FGD metric, enabling pairwise comparisons between the gesture set from the GENEA test audio data and each of the gesture sets from the audio datasets with added noise at different levels. The results of the comparisons for both FGD and MSE are provided in Table 1 right part in Section 4.

## 4 Results and Discussion

The robustness evaluation results for the DiffuseStyleGesture+ model are presented in Table 1. This table displays the FGD and MSE metrics obtained by applying voice conversion (VC) to the audio from the TWH dataset using four different speaker voices from the ESD dataset, as well as for audio with different levels of added noise (low, medium, and high) the THW-party.

The table shows that the model performed best under the Man High Pitch, with an FGD of 15.74 and an MSE of 2.38. This indicates that the model generated gestures closer to the Speaker 1 when the converted voice had the highest pitch among the male speakers. The Woman Low Pitch followed with an FGD of 20.36 and an MSE of 2.51, suggesting that the model synthesized gestures reasonably well with this voice.

However, the model faced significant challenges under the Woman High Pitch and Man Low Pitch, with FGDs of 28.08 and 50.25, and MSEs of 2.61 and 3.06, respectively. This suggests that extreme pitch variations, both high and low, particularly for female voices and very low-pitched male voices, represent a challenge for the model. These results indicate a decrease in the model's ability to generalize to voices with significantly different pitch ranges.

For noise conditions, the table indicates that the model is quite robust to low levels of noise, with an FGD of 10.56 and an MSE of 1.82. This suggests that the model can maintain gesture accuracy even with some background noise, which is common in real-world applications. As the noise level increases, the model's performance decreases. For medium noise, the FGD rises to 14.23 and the MSE increases to 1.95, showing a moderate impact on the quality of gesture generation. In high noise conditions, the FGD reaches 17.59 and the MSE 2.01, indicating a more substantial deterioration in performance.

Overall, the results demonstrate that DiffuseStyleGesture+ exhibits a certain degree of robustness to unseen voices and environmental noise. The model performs better with voices that approximate the original pitch of Speaker 1 in the TWH dataset and with lower levels of noise. This highlights the importance of handling noise in speech-driven gesture generation models. Although the DiffuseStyleGesture+ model shows some resilience to environmental noise, its robustness is compromised at higher noise levels. This demonstrates the need for further development to enhance noise robustness, ensuring reliable performance across a variety of real-world acoustic scenarios. In particular, extreme variations in pitch, both high and low, appear to be a challenge for the model. Future improvements should focus on expanding the range of voices and acoustic conditions that the model can effectively handle.

## 5 Conclusion

This work evaluated the robustness of speech-driven gesture generation models, focusing on generalization to unseen voices and resistance to environmental noise. We used a voice conversion model to generate synthesized audio and created a controlled dataset with background noise, named TWH-party. The evaluation was conducted using the DiffuseStyleGesture+ model. Results show that DiffuseStyleGesture+ model exhibits robustness to unseen voices and noise, performing best with voices similar in pitch to Speaker 1 in the TWH dataset and with lower noise levels. However, performance decreases as noise level and pitch variation increase.

This work contributed to establishing a framework for systematically evaluating gesture generation model robustness, providing a method and dataset for assessing generalization to new voices and noisy environments.

Limitations include using a single gesture generation model. As Future work should evaluate more speech-driven models and use larger datasets like VCTK, which contains 109 voices. Additionally, we aim to expand the proposed robustness protocol to unseen languages.

## Acknowledgments

This study was partially funded by the Coordenação de Aperfeiçoamento de Pessoal de Nivel Superior – Brasil (CAPES) – Finance Code 001. This project was supported by the Ministry of Science, Technology, and Innovations, with resources from Law No. 8.248, of October 23, 1991, under the PPI-SOFTEX program, coordinated by Softex and published as Cognitive Architecture (Phase 3), DOU 01245.003479/2024-10; by the São Paulo Research Foundation (FAPESP) under grant 2020/09838-0 (BI0S - Brazilian Institute of Data Science); by the Eldorado Research Institute and by the Artificial Intelligence Lab., Recod.ai.

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
