# OpenReview forum: "Benchmarking Speech-Driven Gesture Generation Models for Generalization to Unseen Voices and Noisy Environments"
_ACM.org/ICMI/2024/Workshop/GENEA — GENEA Workshop 2024_

### Official Review · Reviewer_L64G · 2024-07-26
**This paper evaluates the robustness of a speech-driven gesture generation model "DiffuseStyleGesture+", in terms of input pitch range and noise environments. Objective measures using FGD and MSE metrics are conducted, but subjective evaluation is lacking.**

**Rating:** 6
**Confidence:** 4

**Review:**

This paper evaluates the robustness of a speech-driven gesture generation model "DiffuseStyleGesture+", in terms of input pitch range and noise environments. Objective measures using FGD and MSE metrics are conducted, but subjective evaluation is lacking.

Regarding pitch range, in general, higher pitch voices are usually accompanied by larger and faster gesture movements, while lower pitch voices are usually accompanied by smaller and slower gesture movements, for a specific individual. So, it can be expected that you will find discrepancies between the gestures generated by different pitch range voices. In this sense, it is questionable if the objective measures are appropriate to evaluate the robustness of the models, so that subjective evaluation should be added to support the conclusions.

I understand voice conversion technologies are useful for data augmentation, but it is questionable if the same gestures can be associated to the converted voices. It would be worth to conduct subjective experiments also to check if this association is acceptable.

Regarding the robustness to environmental noises, it would be more appropriate to quantify noise levels in terms of SNR (Signal-to-Noise Ratio). (I guess S = -3dB represents the attenuation level from the original level, while we don't know what is the noise level. I also would recommend to test SNR levels in 3dB intervals. The perceptual differences of 1dB intervals are small.)

**Nominate For A Reproducibility Award:**

no

---

### Official Review · Reviewer_rPVv · 2024-07-26
**The paper "Benchmarking Speech-Driven Gesture Generation Models for Generalization to Unseen Voices and Noisy Environments" addresses the challenge of generalizing speech-driven gesture generation models to new voices and noisy environments. It proposes an evaluation method using augmented data produced with voice conversion model and introduces a synthetic noisy dataset for evaluation. The findings indicate performance degradation in the DiffuseStyleGesture+ model under various voices and noise conditions, highlighting areas for improvement in gesture generation models.**

**Rating:** 6
**Confidence:** 4

**Review:**

Strengths:
Addresses an important and relevant problem in the field.
Proposes a novel and simple data augmentation for co-speech gesture generation.
Experimental setup and results analysis.
Weaknesses:
Lacks sufficient references and supporting evidence in some sections.
Needs more clarity and structure in explanations.
Should evaluate a broader range of models to generalize findings.


Detailed Review
1. Introduction
References: The introduction needs more references, especially in the last sentence of the first paragraph. Including additional citations would provide a stronger foundation and show how this study builds on existing work.
Discussion on Similar Datasets: To help readers understand the significance of the proposed dataset, It would be helpful to discuss similar datasets, with advantages and disadvantages, and justify the need for augmenting the data.
Gestural Differences in Environments: The paper could explore how gestures vary in different environments (e.g., a fancy restaurant vs. a club with loud music).  Also, differences in gender and emotion significantly impact body gestures. This aspect should be discussed.

2. Related Work
Gesture Datasets in Different Environments: The paper should address whether there are existing gesture datasets from different environments. If such datasets exist, discussing them could highlight gaps that the current work aims to fill. Extracting data from public videos could be a viable suggestion for future work.

2.1 Methodology
Paragraph 2 Support and References: The second paragraph starting at line 142 needs more supporting references. If the approach has been successful in singing, it should be backed with citations. Also, discussing whether this success translates to regular conversational speech requires more evidence.
Clarity and Notation: Section 2.2 could be clearer. Improving notation and providing a more detailed explanation of the process could increase the clarity.

2.2 Metrics
This section explaining FGD and MSE in details which seems redundant. These metrics are well-known in the field. A brief description in a few sentences should suffice. Emphasizing that these are standard quantitative metrics used in the paper would be more concise.

3. Experimental Setup
The term “untrained condition” is not clearly defined. A more detailed explanation is needed to define ut and how it impacts the results. Is there a trained model with augmented data?
While DiffuseStyleGesture+ is a good choice, evaluating more models to generalize the results is necessary. The paper could briefly mention other models like transformers or GAN-based architectures and compare their performances rather than explaining the DiffuseStyleGEsture+ in details.
Line 240 needs a citation.
Line 242’s claim regarding mel-spectrograms and MFCCs requires either a citation or experimental evidence to support it.

3.2 Experimental Results
The statement on line 268 about expressive data and prosodic variations needs supporting evidence. The paper should provide references or experimental results to back this claim.
Frequency Matching Algorithm: The explanation of the frequency matching algorithm lacks clarity and should include notation to make it more comprehensible.


General Points:
The explanation of the training data and inputs for each step is hard to follow. A more structured presentation of this information would help readers understand the experimental setup better.
The conclusion effectively highlights the contributions of the paper. The proposed future work seems promising and addresses important aspects for further improvement.
Consider including a section on user studies or human evaluations to complement the quantitative metrics. User feedback on the generated gestures could provide insights into the model’s practical applicability.

The paper shows promise but needs improvements in several areas. Add more references and Improve clarity and structure in explanations. Provide stronger support for claims with citations or experimental evidence. Evaluate more models to generalize the results.
With these revisions, the paper could significantly contribute to the field of speech-driven gesture generation. The proposed evaluation framework has the potential to become a standard for robustness testing.

**Nominate For A Reproducibility Award:**

No

---

### Decision · Program_Chairs · 2024-07-31

**Decision:**

Accept

**Comment:**

This short paper evaluates the robustness of DiffuseStyleGesture+ under four new speakers and three levels of artificial noise. The paper computes the FGD and the MSE of the generated outputs in each setting, showing that the quality of the model’s outputs highly depend on which voice is used. In comparison, the performance degradation between increasing noise levels is less severe.

Reviewer rPVv raised concerns about missing references and details, while Reviewer L64G underlines the importance of incorporating subjective evaluation into the methodology. Additionally, some details are missing regarding the FGD autoencoder implementation (e.g., is it trained on frames or sequences?), and the experiments should include noise-free and known-speaker conditions for reference.

Overall, I agree with the reviewers’ recommendation to accept the paper, as it highlights an important next step for the evaluation of gesture models in the context of real-world deployment. Please make sure that the data and the generation scripts are published (as claimed in the paper) when the camera-ready version is submitted - the provided link currently only seems to contain sample files.